# Ethanol Disrupts the Protective Crosstalk Between Macrophages and HBV-Infected Hepatocytes

**DOI:** 10.3390/biom15010057

**Published:** 2025-01-03

**Authors:** Murali Ganesan, Anup S. Pathania, Grace Bybee, Kusum K. Kharbanda, Larisa Y. Poluektova, Natalia A. Osna

**Affiliations:** 1Department of Pharmacology and Experimental Neuroscience, University of Nebraska Medical Center, Omaha, NE 68198, USA; biomurali@gmail.com (M.G.); anup.pathania@unmc.edu (A.S.P.); gbybee@unmc.edu (G.B.); lpoluekt@unmc.edu (L.Y.P.); 2Department of Internal Medicine, University of Nebraska Medical Center, Omaha, NE 68105, USA; kkharbanda@unmc.edu; 3Research Service, Veterans Affairs Nebraska Western Iowa Health Care System, Omaha, NE 68105, USA

**Keywords:** HBV, macrophages, hepatocytes, ethanol, interferon-stimulated genes

## Abstract

About 296 million people worldwide are living with chronic hepatitis B viral (HBV) infection, and outcomes to end-stage liver diseases are potentiated by alcohol. HBV replicates in hepatocytes, but other liver non-parenchymal cells can sense the virus. In this study, we aimed to investigate the regulatory effects of macrophages on HBV marker and interferon-stimulated genes (ISGs) expressions in hepatocytes. This study was performed on HBV-replicating HepG2.2.15 cells and human monocyte-derived macrophages (MDMs). We found that exposure of HepG2.2.15 cells to an acetaldehyde-generating system (AGS) increased HBV RNA, HBV DNA, and cccDNA expressions and suppressed the activation of ISGs, *APOBEC3G*, *ISG15*, and *OAS1*. Supernatants collected from IFNα-activated MDMs decreased HBV marker levels and induced ISG activation in AGS-treated and untreated HepG2.215 cells. These effects were reversed by exposure of MDMs to ethanol and mimicked by treatment with exosome release inhibitor GW4869. We conclude that exosome-mediated crosstalk between IFN-activated macrophages and HBV-replicating hepatocytes plays a protective role via the up-regulation of ISGs and suppression of HBV replication. However, ethanol exposure to macrophages breaks this protection.

## 1. Introduction

Acute hepatitis B virus (HBV) infection is a self-limiting disease with the clearance of HBsAg within six months. In all age groups, about 4.1% of hepatitis B patients develop chronic hepatitis B (HBV) infection, which affects about 257–291 million persons worldwide and is associated with substantial morbidity and mortality as well as outcomes to fibrosis, cirrhosis, and hepatocellular carcinoma (HCC) [1]. Furthermore, the risk of end-stage disease development in HBV patients with alcohol use disorder (AUD) is increased up to 1.5-fold for mild or moderate drinkers and up to 8.4-fold for heavy drinkers [2], leading to enhanced HBV load and immune disbalance [3].

HBV is a DNA virus that replicates in hepatocytes [4]. However, other liver-populating cells, like macrophages (both resident and those from circulation), are also exposed to HBV and related viral products and can be activated by interferons from various sources. This finally affects macrophage properties and modifies the crosstalk between these cells and HBV-infected hepatocytes [5]. The outcome of HBV-mediated macrophage–hepatocyte cross-communication depends on the stage of the disease and the prolongation of macrophage exposure to HBV. Multiple environmental exposures, including alcohol, regulate it. Both alcohol and HBV induce metabolic changes in the liver, which, in turn, may cause intrahepatic inflammation [6].

We hypothesize that short-term crosstalk between HBV-replicating hepatocytes and activated macrophages may control HBV levels in hepatocytes. Their crosstalk occurs via extracellular vesicles (EVs) and up-regulates anti-viral interferon-stimulated gene (ISG) levels in hepatocytes to resist HBV. However, this feedback protection is disrupted by ethanol exposure to macrophages.

## 2. Materials and Methods

### 2.1. Reagents and Media

We purchased high-glucose Dulbecco’s Modified Eagle Medium (DMEM) and fetal bovine serum from Invitrogen (Carlsbad, CA, USA), TRIzol from Life Technologies, primer probes and reverse transcription polymerase chain reaction (RT-PCR) reagents from Applied Biosystems by Thermo Fisher Scientific (Asheville, NC, USA), and other analytical-grade quality reagents from Sigma (St. Louis, MO, USA). Anti-STAT1/STAT1 antibodies were purchased from Cell Signaling (Beverly, MA, USA)

### 2.2. Cells and Treatments

We used HepG2.2.15 cells as a source of HBV-positive cells and HBV-negative HepG2 cells for control. HepG2.2.15 cells were derived from HepG2 cells and demonstrated stable HBV expression and replication in the culture system [7]. They were cultured in Dulbecco’s Modified Eagle Medium (DMEM) supplemented with 10% fetal bovine serum (FBS), 1% penicillin/streptomycin, and 0.8% Geneticin. We have long-term experience working with this cell line for HBV studies [8]. Because significant ethanol effects on HBV replication come from ethanol metabolism [8] and these cells do not express ethanol-metabolizing enzymes, we used an acetaldehyde-generating system (AGS) to mimic ethanol metabolism. Briefly, the AGS contained yeast alcohol dehydrogenase (ADH), 50 mM ethanol as a substrate, and nicotinamide adenine dinucleotide (NAD) as a cofactor and provided continuous enzymatic generation of physiologically relevant amounts of acetaldehyde for at least 72 h. AGS was prepared in serum-free DMEM supplemented with 1% penicillin/streptomycin, using 22 µL of 90 mM NAD, 3 µL of 100% ethanol, and 0.02 units of ADH. AGS has been successfully used in our previous studies, and its effects are comparable with the effects of ethanol in metabolizing primary human hepatocytes (PHH) [8,9,10]. Cells were exposed to AGS for 48 h and then processed for end-point detections (HBV RNA, HBV DNA, cccDNA, Western blotting for proteins, etc.). We also used HepAD38 cells to confirm the effects of AGS on HBV replication. These HBV-replicating cells were purchased from ATCC and maintained according to ATCC protocol. They replicate more HBV with higher levels of cccDNA than HepG2.2.15 cells and are cultured in a tetracycline-free medium; the addition of tetracycline suppresses their ability to express HBV markers. They were cultured in high-glucose DMEM supplemented with 10% tetracycline-free FBS, 1% penicillin/streptomycin, 0.8% Geneticin, and 1% 1M HEPES. 

Human monocyte-derived macrophages (MDMs) were used as the source of macrophages. MDM generation is detailed in [9]. Briefly, monocytes obtained from healthy donor elutriation were cultured in DMEM supplemented with 10% inactivated human serum, 1% glutamine, 50 g/mL gentamicin, and human CSF-1 for seven days. The cells were untreated or treated with 25 mM ethanol or the exosome release inhibitor GW4869, 10 µM for 48 h. For interferon-alpha (IFNα) signaling induction, 200 IU human IFNα was added for four hours to activate interferon-stimulated genes (ISGs).

### 2.3. RNA and DNA Isolation, RT-PCR and ddPCR

All reagents were from Life Technologies and Applied Biosciences, purchased from Thermofisher Scientific (Carlsbad and Foster City, CA, USA). RNA was isolated with TRIzol. Using a high-capacity reverse transcription kit, we reverse-transcribed it to cDNA, following a two-step procedure. Then, cDNA was amplified using TaqMan Universal Master Mix II with fluorescently labeled primers, and the relative quantity of each RNA transcript was calculated by its threshold cycle (CT) after subtraction of the reference cDNA (GAPDH). HBV infection was confirmed by measuring HBV RNA. HBV DNA was quantified by digital droplet PCR (ddPCR). Briefly, the 20 µL ddPCR reaction comprised 2X ddPCR Supermix (5 µL), reverse transcriptase (2 µL), 300 mM DTT (1 µL; Bio-Rad, Pleasanton, CA, USA), 900 nmol/HBV sense (5’GA CGT GCA GAG GTG AAG3’) and antisense (5’CAC CTC TCT TTA CGC GGA CT-3’) primers, 250 nmol HBV probe (5’-/56-FAM/ATC TGC CGG/ZEN/ACC GTG TGC AC/3IABkFQ/-3’), and 5 µL of adjusted DNA sample in RNase-free water. Primers and probes were from Integrated DNA Technologies (Coralville, IA, USA). The droplets were transferred to a Bio-Rad 96-well PCR plate using an Automated Droplet Generator. The PCR plate was heat-sealed with pierceable foil using the PX1 PCR plate sealer (Bio-Rad, Hercules, CA, USA) and amplified in the C1000 Touch deep-well thermal cycler (Bio-Rad). The cycling conditions were as follows: an initial denaturation cycle of 10 min at 95 °C, followed by 45 cycles of denaturation for 30 s at 94 °C, annealing for 60 s at 57 °C (ramping rate set to 2 °C/s), and a final incubation for 10 min at 98 °C, ending at 4 °C. After amplification, the 96-well plate was fixed in a plate holder and placed in the QX200 Droplet Reader (Bio-Rad, Hercules, CA, USA). The ddPCR data were analyzed using QuantaSoft analysis software (Bio-Rad, Hercules, CA, USA).

### 2.4. HBsAg Sandwich ELISA

HBsAg was measured in the cell lysates with a quantitative ELISA kit (LifeSpan Biosciences, Seattle, WA, USA).

### 2.5. Western Blot/Immunoblotting and Immunoprecipitation

Western blot has been used to measure the pSTAT1/STAT1 ratio as described elsewhere [11]. Briefly, we prepared cell lysates in 0.5 M EDTA, 2M Tris (pH 7.2), 20 mM Na_3_VO_4_, 200 mM Na_4_P2O_7_, 100 mM PMSF, 1 M NaF, 20% Triton X-100, and aprotinin, pH 7, and nuclear fractions were collected by the kit (Active Motif, Carlsbad, CA, USA). Immunoblotting was developed using the Odyssey infrared imaging system. Protein bands were quantified by Li-Cor software (version 2.2, Linkoln, NE, USA). As a loading control for proteins, β-actin has been used. For phosphorylated STAT1 (pSTAT1), the ratio between pSTAT1 and total STAT1 was calculated. For immunoprecipitation, antigen–antibody complexes were incubated with protein G Sepharose (GE Healthcare Biosciences, Uppsala, Sweden) overnight at 4 °C then washed and incubated with SDS-PAGE sample-solubilizing buffer at 95 °C for 10 min. Isotype-specific IgG was used as a negative control.

### 2.6. In Vivo Studies on HBV Transgenic Mice (Tg05)

HBV-replicating mice were obtained from J.H. James Ou, University of South California. They were maintained in a pathogen-free animal facility, and the study was approved by the Institutional Animal Care and Use Committee of the University of Nebraska Medical Center. They were fed either control or ethanol liquid diets for 10 days then gavaged and sacrificed in 9 h [12]. We used 5 mice in the control and 5 mice in the ethanol group. 

### 2.7. Statistical Analysis

Data from three duplicate independent experiments were expressed as mean ± SEM and analyzed with an unpaired t-test with Welsh correction. Comparisons between multiple groups were performed using ANOVA (Tukey’s post hoc test). Comparisons between the two groups were made using the Student’s test. A probability value equal to 0.05 or less was considered significant. 

## 3. Results

### 3.1. Acetaldehyde Induces HBV Markers Expression and Suppresses Interferon-Stimulated Genes (ISGs) Activation in HepG2.2.15 Cells

We studied how acetaldehyde constantly generated by AGS affected the expression of HBV DNA and HBV RNA in HBV-replicating HepG2.2.15 cells and HBsAg levels in cell supernatants. In this regard, cells were exposed to AGS for 48 h. HBV DNA was measured by ddPCR, HBV RNA by RT PCR, and ELISA quantified HBsAg in cell supernatants. We found that AGS treatment up-regulated HBV DNA by 2.5-fold, HBV RNA by 2-fold, and the release of HBsAg in cell supernatants by 3-fold (Figure 1).

We also used HepAD38 cells to confirm that the effects of acetaldehyde on HBV markers are not restricted to the HepG2.2.15 cell line. These cells were seeded at the concentration of 1 × 10^6^/25 cm^2^ flask and left untreated or treated with AGS in tightly closed flasks for 48 h (Figure 2). Then, we measured HBV RNA, HBV DNA, and cccDNA as specified for HepG2.2.15 cells. We observed 1.5-fold induction of HBV RNA, about 6-fold induction of HBV DNA, and about 4-fold induction of cccDNA by AGS in these cells. Thus, the up-regulating effects of AGS on HBV replication were confirmed on another cell line.

Next, we measured the effects of AGS on the induction of anti-viral ISGs, 2’-5’-oligoadenylate synthetase 1 (*OAS1*), apolipoprotein B mRNA editing enzyme, catalytic subunit 3G (*APOBEC 3G*), and interferon-stimulated gene 15 (*ISG15*) by IFNα in HepG2.2.15 cells. The products of these genes demonstrated essential antiviral activity in many infections.

Cells were left untreated or exposed to AGS for 48 h and treated with recombinant human IFNα, 200 IU, 4 h before harvesting. Then, HBV RNA was purified, and ISGs were detected using RT-PCR. It appeared that ISG activation was decreased by AGS exposure (Figure 3A–C). To measure cccDNA, we used ddPCR. We compared the effects of AGS on cccDNA and *APOBEC3G*, *ISG15*, and *OAS1* expressions in HepG2.2.15 cells exposed to IFNα.

We found that while AGS exposure to cells suppressed *APOBEC3G* and other ISGs induction by IFNα, it increased cccDNA levels (Figure 3A–D). Notably, cccDNA expression was not affected by IFNα treatment (Figure 3D). 

### 3.2. In Vivo Effects of Ethanol on HBV Replication and Ex Vivo ISG Activation in Hepatocytes

Hepatocytes were obtained from the livers of TG05 HBV-replicating mice by collagenase perfusion. Then, they were plated on collagen to measure HBV RNA and HBV DNA (Figure 4A,B). We observed the induction of HBV RNA and DNA by ethanol feeding to mice. Activation of ISGs was induced by mouse IFNα, 200 IU, applied for 4 h. ISGs’ mRNAs were suppressed in hepatocytes isolated from ethanol-fed mice (Figure 4C).

### 3.3. Ethanol Metabolism Impairs IFNα Signaling in HepG2.2.15 Cells

To determine whether the decrease in ISG activation is due to AGS-impaired IFNα signaling in HepG2.2.15 cells, STAT1 phosphorylation was induced by exposure of untreated and AGS-treated cells to IFNα (IFNα, 1000 IU for 30 min). STAT1 has been measured in IFNα-non-treated cells. Phosphorylated STAT1 (pSTAT1) and total STAT1 were measured using an immunoblot (IB). As shown in Figure 5, the pSTAT1/STAT1 ratio was suppressed by AGS exposure about 3-fold, indicating that AGS suppressed interferon signaling at the upstream levels of STAT1 phosphorylation.

### 3.4. Ethanol Exposure to MDMs Attenuates the Activation of Anti-Viral ISGs

The next question we asked was whether the expression of antiviral ISGs is affected by ethanol treatment of macrophages. Here, we used ethanol instead of AGS because, unlike HepG2.2.15 and HepAD38 cells, macrophages can metabolize ethanol. After human monocytes were differentiated into macrophages for 7–8 days, MDMs were treated or not treated with 25 mM ethanol (EtOH) for 48 h, and ISGs were induced by IFNα, 200 IU, for 4 h. ISG expression was measured by RT-PCR. EtOH attenuated *APOBEC3G*, *ISG15*, and *OAS1* gene activation (Figure 6).

### 3.5. Ethanol Exposure to MDMs Attenuates IFNα Signaling

The next step was to test whether ethanol attenuates IFNα-induced signaling in MDMs as with AGS in HepG2.2.15 cells. Thus, MDMs were exposed to 1000 IU for 30 min to induce STAT1 phosphorylation, and the pSTAT1/STAT1 ratio was determined by immunoblot. However, in our hands, exposure of MDMs to ethanol did not affect the pSTAT1/STAT1 ratio (Figure 7A). So, although ISG activation in MDMs was compromised by ethanol exposure, ethanol did not suppress the upstream level of IFNα signaling. Since STAT1 methylation is required for attachment of activated STAT1 to DNA (to activate ISGs) [13,14], we tested STAT1 methylation to characterize the downstream level of IFNα signaling in MDMs. For this purpose, we performed immunoprecipitation with an antibody to methyl arginine and then probed the blot with an antibody to STAT1. Exposure of MDMs to ethanol inhibited STAT1 methylation, a prerequisite for impaired attachment of the STAT1–STAT2–IRF9 complex to DNA required for activating anti-viral ISGs (Figure 7B).

### 3.6. Regulation of HBV RNA and Expression of ISGs in HepG2.2.15 Cells by MDM Supernatants

Here, we tested whether supernatants derived from MDMs treated or untreated with EtOH and IFNα regulated HBV RNA and ISG mRNA levels in HepG2.2.15 cells (marked as HBV). To prove the involvement of EVs, MDMs were treated with the inhibitor GW4869 (10 μM), which blocks exosome release. MDMs were pre-treated with 25 mM EtOH or control medium (48 h) in the presence or absence of IFNα (200 IU, for 4 h). Conditioned MDM medium was collected and applied to HepG2.2.15 cells for 48 h to measure HBV RNA and APOBEC3G and ISG15 mRNA levels by RT-PCR. We found that supernatants collected from MDMs treated with IFNα suppressed HBV RNA expression. In contrast, medium from ethanol-treated MDMs exposed to IFNα did not affect HBV RNA expression. Likewise, activation of *APOBEC3G* and *ISG15* genes in HepG2.2.15 cells was induced by medium from IFNα-activated MDM but not by medium from MDM exposed to the combination of IFNα and ethanol. These effects of ethanol were mimicked by exposure to the exosome inhibitor GW4869, suggesting exosome involvement (Figure 8). Significantly, based on our long-term experience working with HepG2.2.15 cells in alcohol experiments, 25 mM ethanol neither affects HBV markers nor ISG activation in HepG2.2.15 cells because these cells do not express ethanol-metabolizing enzymes and, thus, cannot generate sufficient levels of ethanol metabolites. So, the reduced protection of HepG2.2.15 cells from the virus by supernatants derived from ethanol-treated MDMs is not attributed to the direct effects of leftover ethanol in MDM media.

### 3.7. MDM’s Exposure to Ethanol Disrupts Macrophage-Mediated Protection of HepG2.2.15 Cells Treated with AGS

These experiments were designed to investigate whether ISG activation in HepG2.2.15 cells is regulated by condition medium from MDM and protects either untreated or AGS-treated HepG2.2.15 cells against HBV. To this end, HepG2.2.15 cells (marked as HBV in Figure 8) were exposed or not exposed to AGS for 48 h and then to IFNα, 200 IU, for 4 h to activate ISGs, *APOBEC3G*, and *ISG15*. Additionally, we tested the effects of supernatants collected from IFNα-activated MDMs treated with exosome inhibitor GW4869. As shown in Figure 8, AGS increases HBV RNA levels in HepG2.2.15 cells and attenuates activation of *APOBEC3G* and *ISG15* genes, which was reversed by exposure to a medium collected from IFNα-activated MDMs. We observed no protective effects if the MDM medium was from IFNα non-activated MDMs, EtOH-treated IFNα-exposed MDMs, or MDMs exposed to GW4869 (Figure 9).

## 4. Discussion

This study represents our first attempt to investigate the role of crosstalk between liver-populating circulating macrophages and HBV-replicating hepatocytes with IFNα-activated signaling induced in alcohol exposure settings. As known, HBV may be recognized by toll-like receptor (TLR) 3, RIG/MDA5, or some cytosolic DNA sensors on macrophages. [5]. However, macrophages are not virus-replicating cells, and there is no new formation of HBV RNA after capturing the virus by these cells [15]. Still, all HBV-induced changes are related to transient alterations in cytokine status [16], and IFN type I is considered a macrophage survival factor [17]. Macrophage exposure to external IFNs during viral infections promotes the activation of protective ISGs, which may regulate HBV infection levels in hepatocytes. This setup mimics the situation in acute HBV infection when, in addition to eliminating infected hepatocytes by the adaptive immune response, liver non-parenchymal cells may contribute to the reduction of HBV-specific markers in hepatocytes by modulation of innate immunity in these cells [16,18]. While this is the case for short-term exposure of macrophages to HBV, it may not work in chronic infections when prolonged exposure to HBV significantly alters macrophage phenotype and functions [19,20]. Multiple factors, including EVs, mediate the crosstalk between macrophages and hepatocytes. As demonstrated, HBV RNA and HBV DNA released from HBV+ hepatocytes with exosomes activate immune cells and modulate the host immune response [21,22]. Furthermore, EVs released by activated macrophages may contain ISGs, potentially increasing anti-viral protection [23]. Although the macrophage medium protects HBV-infected hepatocytes [24], it is unclear whether alcohol exposure affects this protection since alcohol strongly influences macrophage properties [25,26]. This is a clinically relevant aspect of HBV infection pathogenesis because by increasing HBV replication and oxidative stress and weakening the immune response, alcohol as a second hit prolongs HBV persistence, serving as a primary trigger of HBV-induced liver cirrhosis [27]. 

This study evaluates the importance of alcohol metabolism for ISG-controlled HBV infection in hepatocytes and the role of alcohol exposure to macrophages in crosstalk between macrophages and HBV-replicating hepatocytes. We have found that in HepG2.2.15 cells, acetaldehyde, an alcohol metabolite, increases HBV RNA, HBV DNA, and cccDNA expressions and suppresses ISG activation by external IFNα. A similar AGS-mediated increase in HBV replication was observed on another HBV-expressing cell line, HepAD38 cells. Notably, a very low dose of IFNα that induces signaling via the JAK-STAT1/STAT2 pathways but does not reduce HBV replication has been used. We cannot exclude the attenuation of ISG activation by acetaldehyde, which is crucial for viral load in hepatocytes [28] and may serve as one of the possible mechanisms for how ethanol metabolism up-regulates HBV marker levels in liver cells. A reverse relationship between HBV cccDNA copy numbers and relative quantities of *APOBEC3* expression has been observed in AGS-treated cells. In fact, in addition to *APOBEG3G* regulating HBV replication, the *APOBEC3* family includes cytidine deaminases that destroy cccDNA [29]. Addressing the mechanism by which ISG activation is suppressed by ethanol metabolism, we focused on the effects of acetaldehyde continuously generated by AGS on IFN type 1 signaling. Previously, we have shown that acetaldehyde suppresses ISGs in hepatocytes due to impaired IFNα signaling via the JAK-STAT1/STAT2 pathway in HCV+ hepatocytes [11]. However, whether this is also the case for HBV infection was unclear. In the current study, we confirmed that in HBV+ HepG2.2.15 cells, AGS also reduced STAT1 phosphorylation, indicating that IFNα signaling is suppressed at the upstream levels. Since ISG induction is regulated via this signaling pathway, AGS limits ISG activation by IFNα.

In MDMs, the induction of ISGs by IFNα treatment was significantly attenuated by ethanol exposure to these cells. Unlike HepG2.2.15, cells that do not express ethanol-metabolizing enzymes and, thus, require AGS treatment instead of exposure to ethanol, MDMs metabolize ethanol to generate reactive oxygen species and acetaldehyde [26] and can be treated with ethanol. To identify the downstream effects of the crosstalk between these macrophages and infected hepatocytes, HepG2.2.15 cells were exposed to a conditioned medium from these MDMs. In our hands, MDM’s supernatants attenuated the expression of HBV RNA. They restored ISGs, *APOBEC3G*, and *ISG15* levels in HepG2.2.15 cells when MDMs were treated with IFNα to induce the IFNα signaling and subsequent upregulation of antiviral ISGs. The partial protection was also attributed to HepG2.2.15 cells treated with AGS. However, MDM-mediated protection was disrupted when IFNα-treated MDMs were exposed to ethanol or an exosome release blocker, GW4869. We have shown that ethanol suppressed ISGs in MDMs due to impaired methylation of STAT1 as a downstream regulator of IFNα signaling. We anticipate that macrophage-derived ISG may be responsible for the protective effects and that exosomes deliver these ISGs to HepG2.2.15 cells. Other products/pro-inflammatory cytokine genes released from macrophages cannot be entirely excluded; they can be packaged to exosomes and delivered to HBV-expressing hepatocytes to regulate HBV RNA levels. Undoubtedly, pro-inflammatory cytokines like IL-1β or TNFα may provide antiviral effects in HBV infection [30,31]. However, in our experiments, the protective effect on HBV replication in hepatocytes has been found only for supernatants from IFNα-treated macrophages, and exposure of hepatocytes to the same dose of IFNα did not affect HBV RNA levels, suggesting that this protection is related to IFN-induced factors, which might be ISGs. As we observed in other experiments, IFNα exposure to macrophages neither affected nor reduced pro-inflammatory cytokine gene expression in these cells. We do not know which component of MDM-secreted exosome cargo is accountable for these antiviral effects. Based on our results demonstrating the reciprocal regulation of cccDNA and *APOPBEC3G* and on previously published data showing *APOBEC3A* or *APOBEC3B* as essential factors for cccDNA degradation [27], we would assume that it may be macrophage-derived *APOBEC3G*, of which suppression occurs under ethanol exposure. Furthermore, in some other infections, like HIV, APOBEC3 proteins were shown to be packaged in exosomes [28], but these studies have not been performed on HBV infection. This matches our results showing the prevention of the regulation of HBV RNA levels by medium derived from IFNα-stimulated MDMs in the presence of exosome release inhibitor GW4869, indicating EV involvement in the delivery of IFNα-activated factors/genes from macrophages to hepatocytes. We believe that we deal with gene, but not protein, delivery because MDMs were exposed to IFNα only for four hours, which, based on our prior experience, is not the optimal time required to increase protein levels. In future studies, we plan to address this issue by analyzing MDM exosome cargo and the role of ISGs in these exosomes in regulating HBV infection in hepatocytes. We will also perform validation experiments. EVs are indispensable tools for liver cell-to-cell communications in infectious and alcohol-related liver disease [32].

## 5. Conclusions

We conclude that by suppressing IFN type 1 signaling, ethanol metabolism attenuates ISG activation in HBV-replicating HepG2.2.15 cells, thereby increasing the viral load in these cells. Medium from short-term IFNα-exposed macrophages may partially reverse the suppression of ISGs in hepatocytes and downregulate HBV marker expression in these HBV-replicating hepatocytes, likely by currently unidentified components of exosome cargo; however, ethanol exposure to macrophages blocks this protection (Figure 10).

## Figures and Tables

**Figure 1 biomolecules-15-00057-f001:**
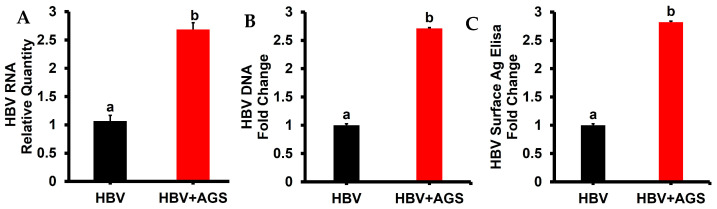
Effects of acetaldehyde on HBV markers. The experiments were performed on HBV-replicating HepG2.2.15 cells exposed or not to AGS for 48 h, and HBV markers were measured. (**A**) Relative quantity of HBV RNA. (**B**) HBV DNA fold change. (**C**) HBsAg concentrations fold change. Values not sharing a common letter are statistically different (*p* < 0.05).

**Figure 2 biomolecules-15-00057-f002:**
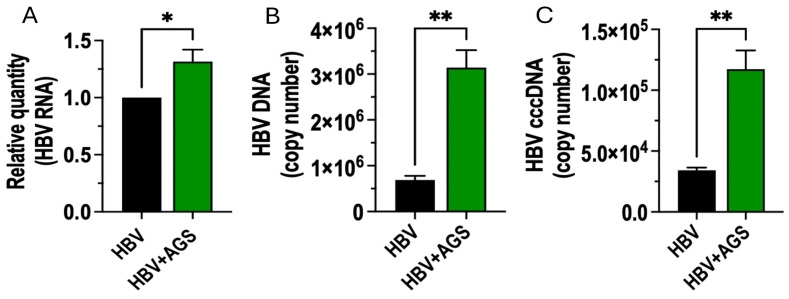
The effects of acetaldehyde on HBV marker expressions in HepAD38 cells. The cells were treated or not with AGS for 48 h, and HBV markers were measured. (**A**) HBV RNA, relative quantity; (**B**) HBV DNA, copy number; (**C**) cccDNA, copy number. * *p* < 0.05; ** *p* < 0.01.

**Figure 3 biomolecules-15-00057-f003:**
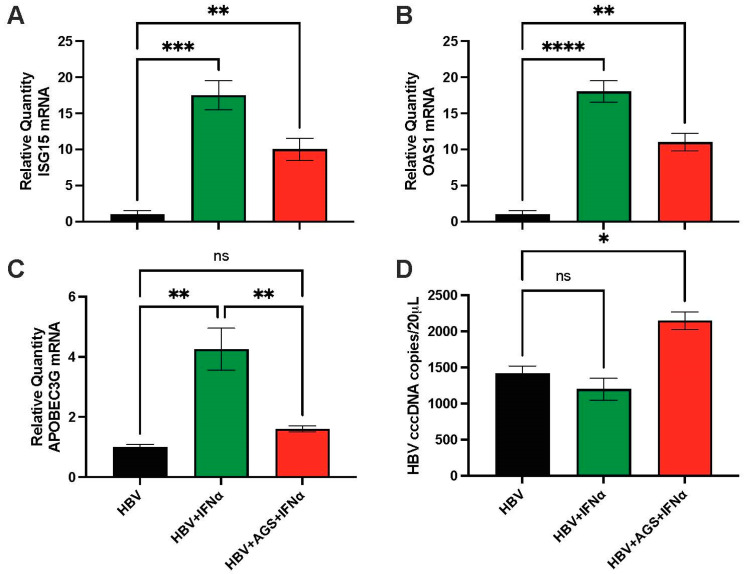
Effect of acetaldehyde on ISG induction and cccDNA levels in HepG2.2.15 cells. The experiments were performed on HepG2.2.15 cells exposed or not to AGS and activated by IFNα (as indicated in the text). (**A**), *ISG15* mRNA; (**B**), *OAS1* mRNA; (**C**), *APOBEC3G* mRNA; (**D**), HBV cccDNA. * *p* < 0.05, ** *p* < 0.01, *** *p* < 0.001, **** *p* < 0.0001, ns-not significant (*p* > 0.05).

**Figure 4 biomolecules-15-00057-f004:**
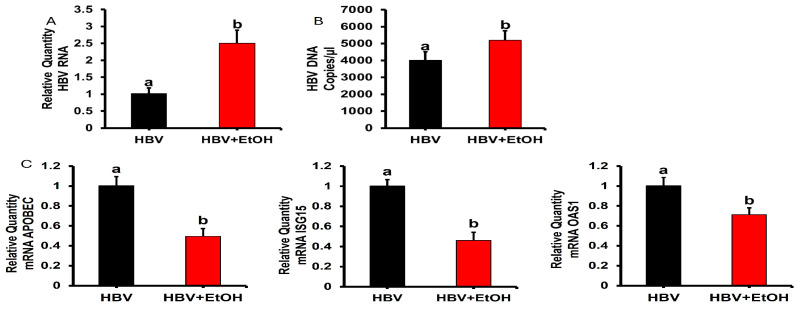
Ethanol feeding to HBV-replicating mice induces increases HBV RNA (**A**) and HBV DNA (**B**) expressions and suppresses induction of ISGs in hepatocytes (**C**). Experiment was designed as indicated in the text. Values not sharing a common letter are statistically different (*p* < 0.05).

**Figure 5 biomolecules-15-00057-f005:**
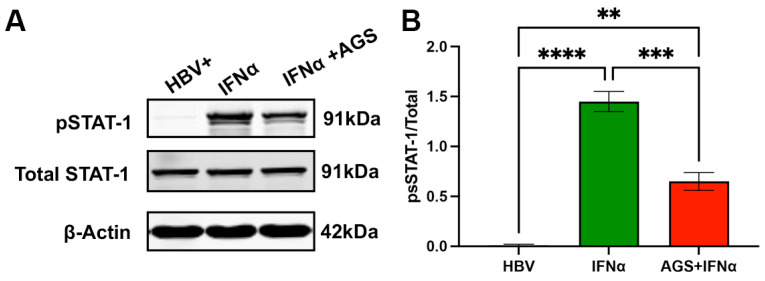
Effects of acetaldehyde on STAT-1 activation. The experiments were performed on HepG2.2.15 (HBV+) cells exposed or not to AGS for 48 h and activated by IFNα (as indicated in the text). Untreated HBV+ cells were used as a control. (**A**) Representative IB. (**B**) Quantitated densitometry of experimental triplicates. ** *p* < 0.01, *** *p* < 0.001, **** *p* < 0.0001, Original images of (**A**) can be found in Appendix A.

**Figure 6 biomolecules-15-00057-f006:**
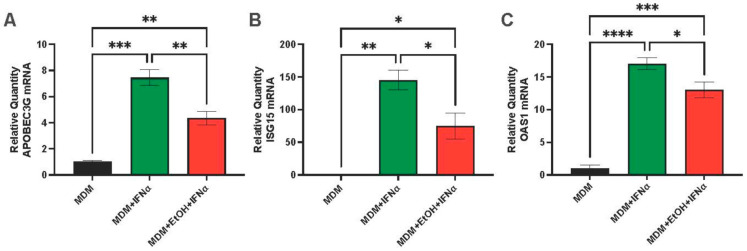
ISG gene expression in MDMs exposed to ethanol. MDMs were left untreated or exposed to ethanol, 25 mM, 48 h and activated by IFNa (200 IU, 4 h). (**A**) Relative quantity of *APOBEC3G* mRNA. (**B**) Relative quantity of *ISG15* mRNA. (**C**) Relative quantity of *OAS1* mRNA. Cells were treated as specified in the text. * *p* < 0.05, ** *p* < 0.01, *** *p* < 0.001, **** *p* < 0.0001.

**Figure 7 biomolecules-15-00057-f007:**
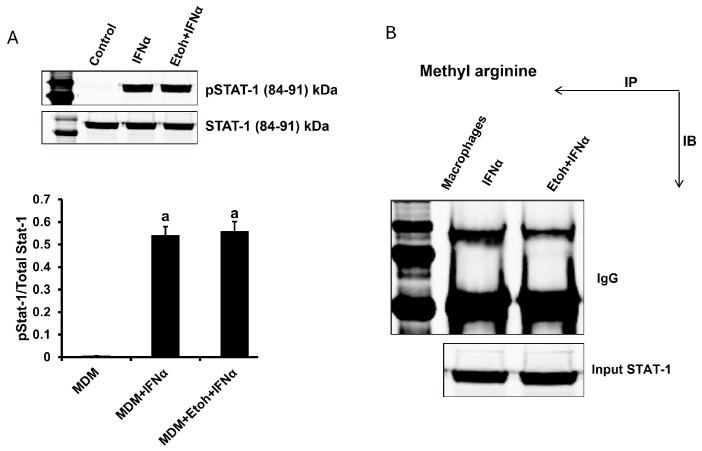
Ethanol exposure attenuates IFNα signaling at the level of STAT1 methylation. MDMs were treated with 25 mM ethanol (EtoH) for 48 h. IFNα, 1000 IU was applied for 30 min to induce the signaling. (**A**) STAT1 phosphorylation (pSTAT1) and STAT1 were quantified by immunoblot (IB), and pSTAT1/STAT1 ratio was calculated. A representative IB image and quantified densitometry of experimental triplicates are presented here. (**B**) STAT1 methylation in MDMs: STAT1 methylation was quantified by immunoprecipitation (IP). IP has been performed by antibody to methyl arginine, and IP bands were probed by anti-STAT1 (IB). The representative image is presented. Original images of (**A,B**) can be found in Appendix A. Values not sharing a common letter are statistically different (*p* < 0.05).

**Figure 8 biomolecules-15-00057-f008:**
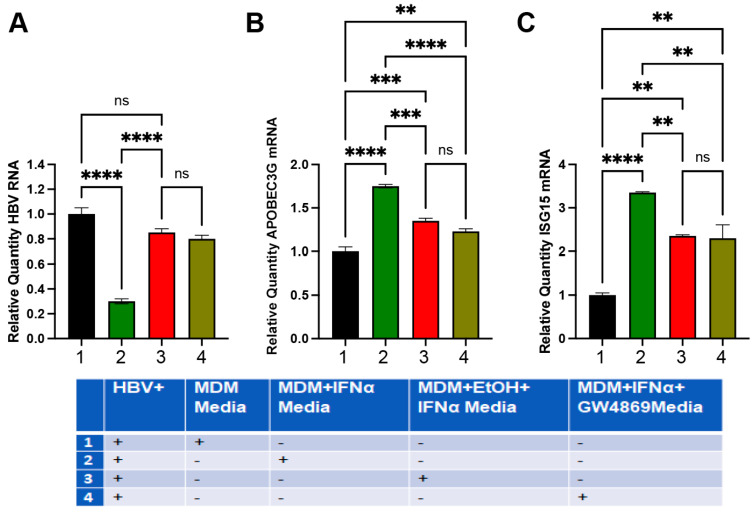
Effects of condition media from MDM exposed or not to ethanol on HBV marker and ISG expression in HepG2.2.15 cells. Experiments were designed as specified in the test. HBV designates HBV+ HepG2.2.15 cells. (**A**) Relative quantity of HBV RNA. (**B**) Relative quantity of APOBEC3G mRNA. (**C**) Relative quantity of ISG15 mRNA. ** *p* < 0.01, *** *p* < 0.001, **** *p* < 0.0001, ns *p* > 0.05.

**Figure 9 biomolecules-15-00057-f009:**
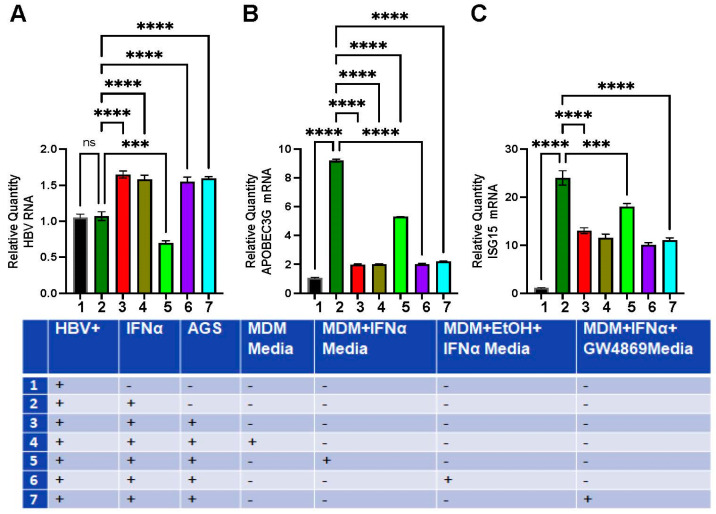
Macrophage-mediated protection of HepG2.2.15 cells is disrupted by EtOH exposure to MDMs. Changes in HBV mRNA content (**A**), APOBEC3G mRNA (**B**), and ISG15 mRNA expressions (**C**). The experiments were designed as described in the text. HBV designates HBV+ HepG2.2.15 cells. *** *p* < 0.001, **** *p* < 0.0001, ns *p* > 0.05.

**Figure 10 biomolecules-15-00057-f010:**
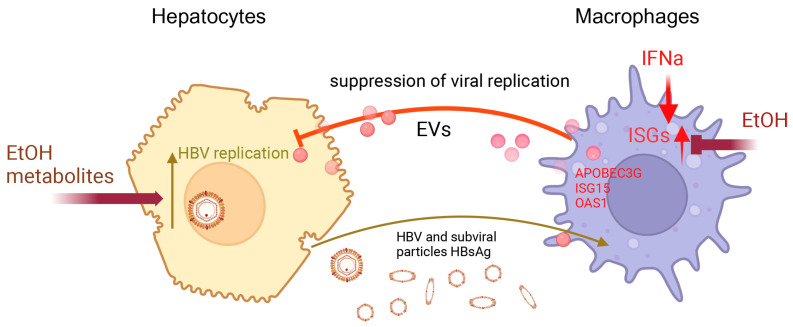
Summary of hepatocyte–macrophage interactions in the context of exposure to ethanol/ethanol metabolites (HBV infection). Exposure to ethanol metabolite acetaldehyde suppresses anti-viral ISGs and increases expression of HBV markers in hepatocytes. In macrophages, IFNα (from internal or external sources) induces ISGs, which may be released with EVs to provide protective effects to HBV-replicating hepatocytes, enhancing ISG expression and reducing HBV replication. This protection is blunted by exposure of macrophages to ethanol due to ethanol-triggered attenuation of IFNα signaling followed by limited activation of ISGs in ethanol-treated macrophages. As a result, these macrophages secrete fewer ISGs in EVs, which are instrumental for regulating HBV replication in hepatocytes.

## Data Availability

Original WB results are provided in Appendix A.

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
