# Peer review of "Ethanol Disrupts the Protective Crosstalk Between Macrophages and HBV-Infected Hepatocytes"

_biomolecules, 2025, doi:10.3390/biom15010057_

Round 1
Reviewer 1 Report
Comments and Suggestions for Authors
Page 3: Line(s)106-108; The temperature of the nucleic acid extension by polymerase enzyme is not mentioned.
Page 3: Line 115-118 Check whether the buffer strengths are correct “Briefly, we prepared cell lysates in 0.5 M EDTA, 2M Tris, 20 mM Na3VO4, 200 mM Na4P2O7, 100 mM PMSF, 1 M NaF, 20% Triton X-100, and aprotinin, pH 7, and nuclear fractions were collected by the kit (Active 117 Motif, Carlsbad, CA).”
Page 5: Line 171: Check the line “HepG2.2.15 cells and found an inversive relationship between these two parameters:”
In Fig 4. “Quantitated densitometry of experimental triplicates” Why “HBV” is written instead of “control”
Comments on the Quality of English LanguageThe manuscript is written very well. The mistakes/modifications have been provided to authors.
Author Response
- Page 3: Line(s)106-108; The temperature of the nucleic acid extension by polymerase enzyme is not mentioned.
Response: In this paper (lines 106-108), we provided the conditions we used for running RT-PCR: The cycling conditions were as follows: an initial denaturation cycle of 10 min at 95°C, followed by 45 cycles of denaturation for 30 s at 94°C, annealing for 60 s at 57°C (ramping rate set to 2°C/s), and a final incubation for 10 min at 98°C, ending at 4°C.
- Page 3: Line 115-118 Check whether the buffer strengths are correct “Briefly, we prepared cell lysates in 0.5 M EDTA, 2M Tris, 20 mM Na3VO4, 200 mM Na4P2O7, 100 mM PMSF, 1 M NaF, 20% Triton X-100, and aprotinin, pH 7, and nuclear fractions were collected by the kit (Active 117 Motif, Carlsbad, CA).
Response: As per request of the reviewer, we re-checked the composition of phosphorylation capture buffer, which was the same as stated in Materials and Methods. We added the pH of Tris buffer to modified version of the paper: 0.5 M EDTA, 2M Tris (pH 7.2), 20 mM Na3VO4, 200 mM Na4P2O7, 100 mM PMSF, 1 M NaF, 20% Triton X-100, and aprotinin, pH 7. We used this buffer in many phosphorylation studies, and it showed the excellent results.
- Page 5: Line 171: Check the line “HepG2.2.15 cells and found an inversive relationship between these two parameters:”
Response: Thank you for this comment. For clarity purposes, we modified this sentence in the following way: We found that while AGS exposure to cells suppressed APOBEC3G and other ISGs induction by IFNα, it increased cccDNA levels (Figure 3 A, B, C, D)
- “Quantitated densitometry of experimental triplicates” Why “HBV” is written instead of “control”
Response: We apologize for this unclarity. We meant that HepG2.2.15 cells are HBV+, and untreated HBV+ cells were used as a control. We clarified this now in the Figure 4 legend: The experiments were performed on HepG2.2.15 (HBV+) cells exposed or not to AGS for 48h and activated by IFNα (as indicated in the text). Untreated HBV+ cells were used as a control. A, Representative IB. B. Quantitated densitometry of experimental triplicates.
Reviewer 2 Report
Comments and Suggestions for Authors
In this manuscript, the authors present studies regarding the interaction between ethanol and interferon response in HBV-infected cell lines and human monocyte-derived macrophages (MDMs). The authors found that interferon response was suppressed in HBV-infected cells following exposure to an acetaldehyde-generating system (AGS). In addition, MDMs seem to protect HBV-replicating hepatocytes, but this protection can be suppressed by ethanol exposure. The study was well designed and the results are interesting.
Question:
1. The viability of HBV-infected cells following exposure to AGS is not shown. Are changes of HBV markers possibly related to cell viability?
Author Response
- The viability of HBV-infected cells following exposure to AGS is not shown. Are changes of HBV markers possibly related to cell viability?
Response: HepG2.2.15 and HepAD38 cells exposed to AGS were viable in our experiments, as evidenced by MTT and LDH assays. In addition, we used load controls for all experiments, so the increase in HBV marker expressions is not due to significant cell death. Recently, we performed experiments on primary human hepatocytes (PHH) treated with 50 mM ethanol (see below), and the effects of ethanol on HBV RNA and DNA were the same as for AGS in HepG2.2.15 and HepAD38 cells.
Reviewer 3 Report
Comments and Suggestions for Authors
The present study investigated the regulatory effects of macrophages on HBV marker and interferon-stimulated genes (ISGs) expressions in hepatocytes using human monocyte-derived macrophages (MDMs) and HBV-replicating HepG2.2.15 cells. The results showed that exosome-mediated crosstalk between IFN-activated macrophages and HBV-replicating hepatocytes plays a protective role via the up-regulation of ISGs and suppression of HBV replication and ethanol exposure to macrophages breaks this protection. The study was overall well conducted and of clinical implications. There are some points that should be further addressed by the authors.
1. Did ethanol exert effects on the ISG gene expression in hepatocytes and macrophages in a dose-dependent way? Treatment with multiple ethanol dose should be considered to address the point.
2. The authors can not conclude that the interactions between hepatocytes and macrophages were mediated by EVs as no assay concerning EVs was conducted.
3. The mechanisms underlying the effect of ethanol remain largely unexplored.
4. The in vitro results should be further validated by in vivo models.
Comments on the Quality of English LanguageThe English could be improved to more clearly express the research.
Author Response
- Did ethanol exert effects on the ISG gene expression in hepatocytes and macrophages in a dose-dependent way? Treatment with multiple ethanol dose should be considered to address the point.
Response: In our in vitro studies, to mimic the in vivo effects of ethanol, we exposed cells to those ethanol concentrations which we were able to reach in our in vivo ethanol feedings to mice and which were reported in the blood of AUD patients. These are 25 to 50 mM ethanol concentrations. As we already mentioned, our HBV-replicating hepatocyte-like cells do not metabolize ethanol and thus, were exposed to AGS. For primary human hepatocytes (PHH), we have chosen 50 mM EtOH since this dose did not provide non-specific toxic effects on these cells (data are presented in response to reviewer 2). For initial experiments, we exposed MDMs to 25, 50, 100 mM EtOH, and there was no difference between exposure to 25mM and 50 mM ethanol in cell viability. For in vitro experiments, we used 25mM ethanol, which corresponds to the concentration of EtOH detected in peripheral blood in vivo and was not toxic for cells. These concentrations are not different from those we already used in other studies and those reported in the literature.
- The authors cannot conclude that the interactions between hepatocytes and macrophages were mediated by EVs as no assay concerning EVs was conducted
Response: That is a very valid comment. However, while we did not report the effects of EVs isolated from macrophages on HBV-replicating HepG2.2.15 cells, we were able to suppress exosome release to supernatants from IFN-activated MDMs by exposure to GW4869, an exosome release inhibitor. GW4869 blunted the protective effect of MDM supernatant on HBV-replicating cells. This allows us to hypothesize that exosome cargo carries out the protection. Currently, we cannot identify the protective component of exosome cargo since this part of the study is in progress. We modified a conclusion by saying: Medium from short-term IFNα-exposed macrophages may partially reverse the suppression of ISGs in hepatocytes and downregulate HBV marker expression in these HBV-replicating hepatocytes, likely, by currently unidentified components of exosome cargo;
- The mechanisms underlying the effect of ethanol remain largely unexplored
Response: The presented data suggest that the mechanism by which ethanol increases HBV replication is related to ethanol-induced suppression of antiviral ISGs. This induction is downstream from IFN signaling, which ethanol metabolism suppresses. This is the case for both hepatocytes and macrophages.
- The in vitro results should be further validated by in vivo models.
Response: Unfortunately, there are no good mouse models to monitor the immune response to HBV, and we recognize this limitation. However, we performed some experiments on HBV-replicating transgenic mice fed control and ethanol diets. The results are presented in 3.2 and Figure 4. We showed that ethanol feeding to mice increased HBV RNA and DNA levels, and ISGs were reduced in hepatocytes isolated from ethanol-fed mice.
Again, we thank the reviewers for their excellent work reviewing the paper and for their valuable comments.
Round 2
Reviewer 3 Report
Comments and Suggestions for Authors
The authors have addressed the concerns and I have no more questions.